# A Review of Effect of Saponins on Ruminal Fermentation, Health and Performance of Ruminants

**DOI:** 10.3390/vetsci10070450

**Published:** 2023-07-10

**Authors:** Ahmed E. Kholif

**Affiliations:** Dairy Science Department, National Research Centre, 33 Bohouth St. Dokki, Giza 12622, Egypt; ae_kholif@live.com or ae.kholif@nrc.sci.eg

**Keywords:** performance, phytogenics, plants, protozoa, saponin, ruminant

## Abstract

**Simple Summary:**

Saponins are active compounds found in plants, with both positive and negative roles in animal nutrition. They are efficient natural rumen modifiers for manipulating ruminal microbial populations, as well as their composition and fermentation. They suppress ruminal ciliate protozoa and may thus enhance microbial protein-synthesis efficiency while abating methane production. The impact of saponins or saponins-containing plants on the ruminal microflora and fermentation depends on the saponin type and level, diet composition, and the microbial community’s composition and adaptation to saponins. Saponins are more effective at enhancing the performance of animals consuming fibrous diets and may be useful to smallholder livestock farmers in developing countries.

**Abstract:**

Saponins are steroid, or triterpene glycoside, compounds found in plants and plant products, mainly legumes. However, some plants containing saponins are toxic. Saponins have both positive and negative roles in animal nutrition. Saponins have been shown to act as membrane-permeabilizing, immunostimulant, hypocholesterolaemic, and defaunating agents in the rumen for the manipulation of ruminal fermentation. Moreover, it has been reported that saponins have impair protein digestion in the gut to interact with cholesterol in the cell membrane, cause cell rupture and selective ruminal protozoa elimination, thus improving N-use efficiency and resulting in a probable increase in ruminant animal performance.

## 1. Introduction

Recently, animal nutritionists and microbiologists are focusing their research interests on the exploration of phytogenics, especially those with selective antimicrobial properties, in the diets of animals to improve rumen metabolism and ruminant performance [1,2,3]. The impetus behind the use of phytogenic feed additives has increased due to the growing demands for organic livestock products. Using phytochemicals to replace antibiotics in animal nutrition is recommended as a result of increasing reservations about antibiotics due to the residues in animal products and the increasing resistance of bacteria to antibiotics [3,4]. Saponins are phytochemicals that have gained increasing interest and are usually administered as a feed additive. 

Saponins are secondary metabolites prevalent in many plants, mainly legumes [5,6,7]. The name saponins’ stems from their ability to form a stable foam in aqueous solutions such as soap [8]. They have the ability to modulate ruminal fermentation and improve animal production [2,7,9]. The US Food and Drug Administration has stated that saponins are ‘Generally Recognized as Safe’ for human consumption [9].

Most legume plants, mainly the leaves and seeds, contain triterpenoid saponins, except *Trigonella foenum-graecum* which contain steroidal saponins. Sapogenin or saponins in plants do not actually occur as a single compound but as many compounds with varied sugar moieties. The roots, leaves, and seeds of alfalfa contain medicagenic acid as the major sapogenin in addition to another 28 saponins [10]. Variations in the sugar moiety, sugar attachment position, and aglycone exert different biological activities [11].

An extract of *Yucca schidigera* has 4.4% steroid saponins with 28 structures of spirostanol and furastonal glycosides [12]. The extract of another plant, *Quillaja saponaria*, contains about 10% saponins with more than 20 varying structures of triterpenoid saponin [12]. The fruits of *Sapindus* have almost the same chemical structure, particularly *Sapindus saponaria* and *Sapindus rarak*, as monodesmoside triterpenoid saponins. Generally, saponins have a high capacity to decrease methane emissions at both low and high levels, indicating the beneficial role of saponins in ruminant nutrition for cleaner environment production [13]. Inhibition of the ruminal methanogen population by saponins was suggested as a mode of action to lower methane emissions [10,13,14].

In the present review, we attempt to provide deeper insights into the effects of saponins or saponins-containing plants on ruminal microbiome and fermentation patterns, saponins metabolism, and how saponins affect ruminant performance.

## 2. Chemistry of Saponins

Chemically, saponins are a group of high-molecular-weight glycosides (1000–1500 Da) containing saccharide chain units (1–8 residues) linked to triterpene saponins (avenacoside) or a steroidal (avenacin) aglycone moiety [2,15,16], with a greater distribution of triterpene types than steroidal types in nature [16]. Figure 1 shows the chemical structures of sapogenins.

There are two major types of triterpenoid-saponins: (1) the neutral type, in which a normal sugar is joined to sapogenin; and (2) the acidic type, in which the sugar moiety consists of uronic acid or one or more carboxylic groups joined to the sapogenin [15,16,17]. Steroid saponins mostly exist in the form of furostanol or spirostanol, where the carbohydrate part contains one or more sugar moieties having glucose, galactose, xylose, arabinose, rhamnose, or glucuronic acid joined to a sapogenin (aglycone) via glycosidic bonds. The number and type of the sugars, and the stereochemistry of the sapogenin moiety, differ greatly, resulting in various groups of metabolites [15]. The saccharide chains are usually joined to the C_3_ position (monodesmosidic), but some sapogenins possess two saccharide chains (bidesmosidic) linked to the C_3_ and C_17_ (via C_28_) positions. Saponins are mostly monodesmosidic or bidesmosidic. Spirostanol is commonly found in monodesmosidic and furostanol bidesmosidic forms [15], where they are principally found in the seeds, roots, and bulbs of plants [15]. Furostanol saponins are located in the assimilatory parts of plants [15]. A mixture of saponins in a single plant species includes cucurbitane, cycloartane, dammarane, holostane, hopane, lanostane, lupane, oleanane, tirucallane, taraxastane, tirucallane, and ursane triterpenoid saponin types [16].

## 3. Occurrence and Roles of Saponins in Plants

Most plants contain saponins in their different parts, such as the root, tuber, bark, leaves, seed, and fruit [15,17]. Plants synthesize saponins as a chemical barrier or shield to enhance their defense system for protecting their tissues. Therefore, they are found at high concentrations in tissues that are most susceptible to pathogen attack or insect predation. Avenacin plays a great role in arresting the zoosporic fungus *Gaeumannomyces graminis*, responsible for the take-all disease, one of the most important wheat root diseases [18].

As previously noted, saponins include mainly triterpene or steroidal saponins. Triterpene saponins and steroidal saponins occur chiefly in either dicotyledons and monocotyledons, respectively, though both can be found in some plant species [9,15]. The concentration of saponins is higher in young leaves than mature leaves and is higher in roots than in foliage [15]. Another type of saponins, soyasapogenols, are mainly found in the axis of seeds axis instead of the cotyledons and seed coat. However, soyasapogenol A is concentrated in the seedlings (the root) and soyasapogenol B is concentrated in the plumule [19].

## 4. Saponins and Rumen Microbiota Population

Patra and Saxena [9], in their review, showed a schematic illustration of how saponins affect rumen microbiota and fermentation. Saponins modify rumen fermentation directly by affecting rumen microbiota (bacteria, fungi, protozoa, and archaea) composition and activity [7,20]. The main determinants of the influence of saponins on rumen microbiota populations are the structure and dose of saponins, animal adaptation, diet composition, and microbial community structure. Different methods of extraction and structures of saponins, may produce diverse effects on rumen metabolism, and may partially explain the differences in response to saponins at the same levels [9,21].

### 4.1. Ciliate Protozoa

Protozoa comprise about 25 to 50% of rumen microbial biomass. Protozoa play diverse roles in feed digestion and H_2_ production [22]. In the rumen, protozoa are divided mainly into two genera: *Holotrichs* and *Entodiniomorphs*, with a preponderance of *Entodiniomorphs*. *Holotrichs* utilize fermentable carbohydrates, while *Entodiniomorphs* degrade starch grains. Protozoa are responsible for about one-fifth of fiber degradation and play a less important role than bacteria and fungi [23]. In ruminants, protozoa engulf and digest ruminal bacteria, fungi, and archaea, and increase ruminal microbial protein turnover, resulting in reduced protein utilization efficiency and increased urea excretion in the urine. *Epidinium ecaudatum*, *Diploplastron affine*, *Eremoplastron bovis*, *Ophryoscolex caudatus*, and *Eudiplodinium maggii* are the most common bacteria engulfers [22]. Moreover, ruminal protozoa have important roles in stabilizing ruminal pH and decreasing the redox potential of rumen digesta. This indirectly promotes the activity of ruminal bacteria.

Ruminal protozoa are very sensitive to saponins and saponins-containing plants or plant extracts (Table 1) [9,10,24,25]. The antiprotozoal activity of saponins is perhaps attributable to the increased permeability of the cell membranes of protozoa, which causes cell contents leakage [9]. Stigmastanol, campestanol, and cholestanol are the common sterols in *Entodiniomorphs*, whereas cholestanol is the chief sterol in *Holotrichs*. The various compositions of ruminal protozoal sterols may explain the variation in ruminal protozoa sensitivity to saponins. Saponins form complexes with sterols in the membrane surface of protozoa and thus affect them by causing impairment and eventual disintegration [9]. Saponins from *Q. saponaria* and tea saponins [26,27] have also shown in vitro antiprotozoal activities. Wallace et al. [28] noted that *Y. schidigera* extracts suppressed ruminal protozoa growth in vitro but did not affect dry matter disappearance and the population of other rumen microbes. In an in vitro experiment [28], protozoal activity, determined by the breakdown of [^14^C] leucine-labelled *Selenomonas ruminantium*, was arrested by 1% *Y. schidigera* extract supplementation, which limited the motion of both the ciliate protozoa and the cilia of *Entodiniomorphs*, and contracted the *Holotrichs*, indicating increased bacteria consumption by protozoa. Widyarini et al. [25] showed a lowered protozoal number with the administration of saponins from *Nigella sativa* L. at 0.2%, 0.4%, and 0.6% DM in vitro. Additionally, Gunun et al. [24] observed lowered numbers of total holotrich, and entodiniomorph protozoa in the rumen of goats fed *Terminalia chebula* at 8, 16, or 24 g per kg DM feed. Hristov et al. [29], using in vitro technique, noted that *Y. schidigera* saponins at 44 to 176 mg and *Q. saponaria* saponins at 100 to 400 mg/L medium did not influence protozoal numbers. Recently, Kim et al. [30] reported a decreased ciliate protozoa number with the administration of saponins-containing *Aloe saponaria* at 37.26 and 45.65 mg/g DM.

Moreover, the in vivo antiprotozoal effects of *Y. schidigera* [40] and *Antidesma thwaitesianum* [7] saponins were previously confirmed in many experiments. Dietary inclusion of ethanol-extracted lucerne saponins (triterpene saponins) at 2 and 4% DM decreased ruminal protozoal counts of sheep by 34 and 66%, respectively [41]. Other experiments showed minimal effects of saponins on ruminal protozoa. In dairy cows, the same effects were observed with the feed *Y. schidigera*, supplying 275 mg saponins/kg DM intake [39]. Gunun et al. [7] observed lowered protozoal counts by feeding lactating cows on a diet supplemented with *A. thwaitesianum* containing saponins at 98 g/kg DM.

Therefore, the impact of saponins on rumen protozoa is dose-dependent, with low saponin levels lacking antiprotozoal effects [31]. A decrease in protozoal counts of between 35 and 40% has been reported in an experiment in which saponins from *Biophytum petersianum* were orally administered at 13 and 19.5 mg/kg BW, respectively, to goats [31]. However, no further reductions were observed when the doses were increased. Many experiments [31] have shown that the optimal doses for the plateaued inhibition of protozoal counts differ among saponin types. The same dose of saponins decreased protozoal numbers by 53% with *Y. schidigera*, by 29% with *Q. saponaria*, and by 40% with *B. petersianum*. These studies suggest that all protozoal species are not equally vulnerable to saponins intoxication, and the potency or efficacy of saponins depends on the saponins type and the protozoal species. Another factor that may affect the antiprotozoal activity is the method by which saponins are extracted from plants [9,21].

Moreover, the impact of saponins on ruminal protozoa depends on diet [14,42]. Research [14,42] has indicated that *S. saponaria* fruits decreased protozoal numbers with *Arachis pintoi*, but had no effect on a medium-quality legume (*Cratylia argentea*). Diet composition has a pronounced effect on the protozoal community, and this diet-dependent effect may be due to the selective activity/effect of saponins on protozoal species.

The lack of an antiprotozoal effect in saponins may be due to the use of low doses, or to rumen microbiota adaptation to saponins. There is a paucity of information in the literature with regards to the influence of saponins on the composition of ruminal protozoa that may have adapted to saponins. Benchaar et al. [39] reported that saponins have no effect on total protozoal populations or on generic composition. It is interesting to note that the antiprotozoal activity of saponins is only transitory [20,43]. The antiprotozoal effects of *Sesbania sesban* in sheep were observed [44,45]. *S. sesban* decreased protozoal counts by 60% after 4 days, but the population recovered after 10 days. Below, the adaptation of rumen microbes to saponins is discussed in detail.

### 4.2. Bacteria

*Fibrobacter succinogenes*, *Ruminococcus albus*, and *Ruminococcus flavefaciens* are the main ruminal plant cell walls that degrade bacterial species, while *Butyrivibrio fibrisolvens*, *Clostridium locheadii*, and *Clostridium longisporum* may be considered as secondary fiber degraders [46]. Many experiments reported that the increased numbers of ruminal bacteria by feeding with saponins [24,30] may be the result of decreased bacteria engulfment by protozoa. Rather than changing the surface tension of the extracellular medium, saponins disrupt the membranes of the microbial cells. The mode of action of saponins on promoting the growth of some bacterial species in pure cultures is unclear. However, cell membrane permeability increased at low doses of saponins in a controlled manner, permitting improved nutrient absorption into bacterial cells [9]. Another explanation for increased bacterial numbers with saponins is the inhibition of protozoal numbers [47].

Wallace et al. [28] showed that an extract of *Y. schidigera* at 1% had no effect on the growth of *S. ruminantium*, stimulated *Prevotella ruminicola*, and inhibited *B. fibrisolvens* and *S. bovis*. In another experiment, Wang et al. [48] showed that both the growth and activity of *R. albus*, *R. flavefaciens*, *S. bovis*, *Prevotella bryantii*, and *Ruminobacter amylophilus* were inhibited, while the growth of *S. ruminantium* was stimulated by *Y. schidigera* saponins. Additionally, Wang et al. [49] noted that the addition of tea saponins to alfalfa hay or soybean hulls fed to Holstein bull-calves for 28 days increased the population of *Prevotellaceae_YAB2003* while it decreased that of *Ruminococcaceae_NK4A214* and *Lachnospiraceae_NK3A20*. Hess et al. [47] observed improved total bacterial counts and reduced total ciliate protozoa counts when lambs were fed grass hay, or a grass and legume mixture, as basal diets supplemented with a concentrate containing crude saponins from *S. saponaria* fruits at 0.6 g/kg BW^0.75^. In other experiments, Liu et al. [35] showed that feeding ewes on a diet supplemented with tea saponins at 2 g/ewe/day enhanced the population of *F. succinogenes*, without affecting total bacteria, methanogen, *R. flavefaciens*, *R. albus*, and *B. fibrisolvens* populations. Additionally, Gunun et al. [24] observed increased bacterial numbers in the rumen of goats fed *T. chebula* at 8, 16, or 24 g per kg DM feed. Kim et al. [30] showed that absolute abundance of general bacteria, *S. ruminantium*, and fungi were improved by the administration of *A. saponaria*. Heat treatment of *A. saponaria* enhanced the abundance of *S. ruminantium*, *P. ruminicola*, and *R. flavefaciences* [30].

The sensitivity of both Gram-positive and Gram-negative bacteria relies on the structure of the aglycone moiety of saponins [50] and the composition of the cell membrane fatty acids of the bacteria under consideration. Gram-positive bacteria are more vulnerable relative to Gram-negative bacteria when subjected to *Y. schidigera* saponins [28]. Avato et al. [50] observed the greater effect of a lucerne saponin, medicagenic acid, against Gram-positive bacteria. Guo et al. [26] showed that tea saponin at 400 mg/L culture medium reduced ruminal fungi populations by 79% but increased *F. succinogenes* numbers by 41% without affecting *R. flavefaciens*.

In another set of experiments, saponins administration showed negative effects or no effects on ruminal bacterial numbers. Wang et al. [48] noted that *Y. schidigera* saponins negatively affected cellulolytic bacteria without any harmful effect on amylolytic bacteria. Muetzel et al. [51] observed that supplementation of saponins-containing *Sesbania pachycarpa* leaves had no effect on the growth of *F. succinogenes* and *R. flavefaciens*; however, *R. albus* was impaired. Gunun et al. [7] observed no effect on bacterial count when cows were fed a diet supplemented with *A. thwaitesianum* as a source of saponins.

The pH of the medium in in vitro experiments and in the rumen of animals in in vivo experiments greatly affects ruminal bacteria sensitivity to saponins administration [9]. Li et al. [52] showed that tea saponins had higher in vitro antimicrobial activity at a low pH, indicating that ruminal pH may modulate the effect of saponins depending on the nature of the diets. Additionally, the effect of saponins on ruminal bacteria is species-dependent, which may allow for a selective manipulation of metabolism in the rumen. The inclusion of saponins in concentrate-based diets inhibited the growth of *S. bovis*, resulting in a reduction in incidences of acidosis [9]. Normally, the population of *S. bovis*, which ferments the soluble starch to lactate, rapidly increases with high-concentrate diets. *Fibrobacter* spp. are obviously more resistant to saponins than other cellulolytic bacterial species because of the presence of 2-aminoethylphosphonic acid in their cell wall, which perhaps improves their membrane stability. 2-aminoethylphosphonic acid, which is covalently joined to the membrane polymers, makes the organism resistant to enzymic hydrolysis and possibly enhances longevity.

### 4.3. Fungi

Anaerobic ruminal fungi have a great role in digesting fiber; however, the proportion of the total ruminal microbial mass is small. Rumen fungi secrete many enzymes that are stronger than those of bacteria. There is little information on the effect of saponins on ruminal fungi. Responses of ruminal fungi to saponins could be mediated through defaunation and/or direct effects. As previously noted, some experiments showed improvements in ruminal fungi populations due to the addition of saponins, and that this attributed to the decline in their engulfment by protozoa. Wang et al. [48] showed the sensitivity of ruminal fungi (*Neocallimastix frontalis* and *Piromyces rhizinflata*) to saponins from *Y. schidigera*. Feeding sheep on diets containing *A. saponaria* [30] increased fungal numbers. Moreover, Wina et al. [53] observed positive effects on ruminal *Chytridiomycetes* fungus in the long-term feeding of low concentrations of saponins from *S. rarak* to sheep but feeding at higher concentrations produced no effect. Wina et al. [54] compared different levels of the methanol extract of *S. rarak* (0, 0.25, 0.5, 1.0, 2.0, and 4.0 mg/mL) in a diet containing elephant grass and wheat bran at 70:30 and observed gradual increases in the microbial biomass as the extract doses increased. Such results indicate that saponins at low doses showed positive effects on rumen fungi. Moreover, they [54] observed a reduced concentration of ruminal fungal RNA in an in vitro fermentation with saponins-containing *S. rarak* extract. However, these effects were not noted in sheep given the same extract of *S. rarak* for three months [53]. Gunun et al. [7] observed no effect on fungal counts in cows fed diet supplemented with *A. thwaitesianum* as a source of saponins.

The antifungal activity of saponins is mainly due to their interaction with sterols in the fungal cells, followed by pore formation and the loss of membrane integrity [9]. Barile et al. [55] showed that saponins could inhibit fungi but not bacteria. They showed that saponins isolated from *Allium minutiflorum* displayed antifungal activity but exhibited no reasonable antibacterial activity in non-rumen bacteria. Saponins with the triterpene or spirostanol moiety normally show stronger antifungal activities, while furostanol saponins with bidesmosidic nature exhibit insignificant or no bacteriostatic and fungicidal effects [56].

### 4.4. Archaea

Although few studies have been conducted on the effects of saponins on ruminal archaea, their effects have gained a lot of attention recently due to their potential for abating enteric greenhouse gases emissions and thus contributing to a cleaner environment. There are more investigations on the quantification of methane production than on the methanogens themselves. In the rumen, about 10–20% of total methanogenic archaea exist in association with protozoa. Therefore, decreasing ruminal protozoal populations could diminish methanogens and methane production [10]. Investigations on pure cultures of ruminal methanogens are limited. Methanogen activity could be directly affected by saponins without any change in methanogen counts, as observed with saponins from *S. saponaria*. Saponins from tea at 400 mg/L did not affect the growth and expression of the methyl coenzyme M reductase subunit A gene (*mcrA*) of *Methanobrevibacter ruminantium*; similarly, in mixed rumen cultures, tea saponins had no effect on total archaeal counts, but decreased *mcrA* gene activity by 76% and methane production by 8% [26]. In another experiment, saponins from *S. rarak* at 4 mg/mL of an in vitro medium reduced methanogen RNA concentrations, without affecting methanogens at a low saponins concentration of <4 mg/mL [54].

In sheep, Hess et al. [47] observed enhanced methanogen counts and decreased methane production with saponins of *S. saponaria* fruits. Recently, Kim et al. [30] observed increased ruminal archaea with *A. saponaria* administration in vitro. Two experiments [57,58] using the RUSITEC system observed that the inclusion of an extract of *Y. schidigera* had no effect on methane production. Lila et al. [59] noted that the addition of sarsaponin extracted from *Y. schidigera* to a starch or a mixed diet decreased in vitro methane emissions.

## 5. Microbial Adaptation and Saponins Metabolism

The inclusion of saponins or saponins-containing plants in diets of animals depresses protozoal count, but the antiprotozoal effect is transient [9,20,43] as the antiprotozoal compounds are degraded by the ruminal bacteria. Saponins from *Y. schidigera* and *S. rarak* fed for 17 d [40] and 3 months [53], respectively, showed antiprotozoal activities in mixed rumen microbial populations. Wina et al. [53] observed that a short feeding period of *S. rarak* saponins showed negative effects on *R. albus*, *R. flavefaciens* and *Chytridiomycetes*, but the effects ceased after a long feeding period. The unaffected protozoal populations in goats fed saponins from *S. saponaria* on day 13 suggest detoxification of the saponins by the ruminal microbes (e.g., bacteria).

The ruminal metabolism of saponins involves deglycosylation [48] and structural alteration of the nucleus of the aglycones. Then, the sapogenins experience structural alterations (oxidation and reduction) less rapidly. *F. succinogenes* effectively deglycosylated saponins from *Y. schidigera* [48]. Increased cell-wall thickness of *P. bryantii* was observed when subjected to *Y. schidigera* saponins, suggesting an adaptation to the saponins [48].

Saponins remain toxic to ruminal protozoa when introduced directly to the rumen due to their higher concentrations and purity. However, the feeding of saponins did not show toxicity signs, suggesting that other factors in the mouths of animals (e.g., chewing, salivary amylase) contribute to detoxification, or that saponins are protected from detoxification/degradation by the feed particles matrix [9]. The wide difference in the degree and rate of ruminal degradation of various saponins indicates the potential of some saponins to remain effective over a long period of time in the rumen. Species, breed, and the environment of the animals are possible contributory factors to the rumen microbe’s ability to mitigate the antiprotozoal activity of saponins. However, the mechanism of adaptation of ruminal microbiota to saponins requires further elucidation. As a way of adaptation, microbes may acquire the ability for rapid degradation of saponins and thus diminish the antiprotozoal activity of saponins. Meagher et al. [60] characterized saponins metabolism in the digestive systems of sheep. According to them, saponins were rapidly hydrolyzed to afford free sapogenins, parts of which passed through oxidation and reduction to produce epismilagenin, smilagenone, smilagenin, and tigogenin, which were assimilated in the duodenum and then released as conjugated and free sapogenins through the bile. Additionally, the epimerization of sapogenins persisted in the caecum and colon.

## 6. Ruminal Degradation of Saponins

The degradation of saponins in the rumen is different to the degradation occurring in other parts of the digestive system. Ruminal bacteria are capable of degrading saponins. However, in an in vitro study, the degradation rate was seen to start slowly and then increase very rapidly after 6 to 8 h of incubation. Meagher et al. [60] observed rapid hydrolysis of saponins (ethanolic extraction of powdered *Costus speciosus* rhizomes followed by butanol–water partitioning) in the sheep of rumen after 1 h of direct introduction. The end products of saponins degradation have not been investigated in detail. An intraruminal infusion of saponins produced other derivative products as a result of saponins degradation. Saponins from *Y. schidigera* and *Narthecium ossifragum* have the same aglycone (sarsapogenin). Besides sarsapogenin, the main ruminal degradation product, the five other derivative products of the sarsapogenin are smilagenin, episarsapogenin, epismilagenin, sarsasapogenone, and smilageno. Nevertheless, a *C. speciosus* rhizomes extract was degraded to aglycone (diosgenin) [60]. Ruminal production of sapogenin and its several derivative products suggests the occurrence of several processes, including ruminal hydrolysis of saponin, epimerization, and hydrogenation of sapogenin [60].

In the gastrointestinal tract, all sapogenins are conveyed throughout the digestive tract and eventually eliminated in the feces. Sapogenins concentrations from the degradation of saponins from *Y. schidigera*, *N. ossifragum* [61], or *C. speciosus* rhizomes [60] were seen in lower concentrations in the duodenum, suggesting duodenal absorption of these products and transportation through the portal vein to the liver where they conjugated with glucuronide and were released into the bile. The free form of sapogenin is not detected in the bile [62]. A post-duodenal increase in all sapogenins concentrations was observed, especially in the caecum and colon [60]. A contentious metabolism of saponins by cecal bacteria was reported [60], with little information available on the specific cecal microorganisms.

## 7. Effects of Saponins on Feed Digestion and Rumen Fermentation

### 7.1. Ruminal Enzyme Activity and Digestion of Feeds

Saponins can affect ruminal bacteria as well as the activity of the ruminal enzymes [25]. In the RUSITEC system, the addition of a *Y. schidigera* extract at 0.5 mg/ml incubation fluid to a complete diet containing alfalfa hay and barley at 1:1 *w*/*w* increased protease activity without affecting the activities of deaminase and peptidase [63]. Widyarini et al. [25] also observed that the administration of saponins at 0.2%, 0.4%, and 0.6% of DM reduced the activity of carboxymethyl cellulase (CMCase) enzymes as saponin levels increased, without affecting amylase and protease activities. Protozoa secrete cellulolytic enzymes, which are responsible for 19–28% of the total cellulolytic activity in the rumen [64]. Decreases in the ruminal activity of xylanase or CMCase appeared to be more related to decreased protozoa than decreased cellulolytic microbes [25].

Saponins may directly inhibit microbial urease, leading to reduced ammonia concentrations in the rumen. Muetzel [51] observed unaffected CMCase activity in an artificial rumen with the administration of saponins-containing *Sophora pachycarpa* leaves as a pure supplement. Additionally, Belanche et al. [21] observed unaffected CMCase activity when 15% DM saponins obtained from Ivy fruit (*Hedera helix*) was administered. However, Hristov et al. [29], using an artificial rumen supplemented with an extract of *Y. schidigera*, showed the decreased activity of CMCase, xylanase, and amylase. In another experiment involving the use of cell-free rumen fluid from steers [40], *Y. schidigera* administration at 60 g/day to steers did not affect CMCase, xylanase, and amylase activities. Sapindus extract at high concentrations (4.0 mg/ml incubation medium) in an artificial rumen markedly reduced xylanase activity [54] as well as in sheep rumen [53]. Saponins were also reported to inhibit the activity of the amylase enzyme [65,66]. Decreased ruminal activity of xylanase or CMCase is more associated with reduced protozoal populations relative to reduced fibrolytic microbes [53].

Ruminal bacteria, protozoa, and fungi are the major lignocellulosic feedstuff degraders in the rumen. Bacteria and fungi contribute to about 80% of the fermentative activity, while protozoa contribute only 20%. The impact of saponins or saponins-containing plants has mixed effects on feed digestion. Saponins exert a substrate-dependent effect on nutrient digestibility, possibly due to their effects on specific bacterial numbers [38]. However, saponins or plants containing saponins had no effect on nutrient digestibility in many experiments [7,36,37]. When supplementing the diet of lactating cows with *A. thwaitesianum* Muell. Arg. pomace containing 98 g/kg saponins, Gunun et al. [7] observed that the digestibility of nutrients was unaffected. The feed amounts of saponins were 9.8, 19.6, and 29.4 g/cow/day.

However, *S. saponaria* [42], *Y. schidigera*, and *Q. saponaria* saponins [33] reduced the in vitro digestibility of fiber (i.e., NDF). Compared to a grass hay diet, Klita et al. [67] observed decreased OM digestibility in sheep fed lucerne saponins at up to 4% of DM intake. The fruits of *S. saponaria* [42] decreased the digestibility of OM and NDF. Wina et al. [53] observed depressed in vitro apparent and true digestibility of the incubated substrate, in a dose-dependent manner. Gunun et al. [24] noted reduced total tract digestibility of CP in goats fed *T. chebula* Retz at 24 g/kg of total DM intake relative to those fed *T. chebula* at 8 or 16 g/kg of total DM intake.

Positive effects of saponins or saponins-containing plants on nutrient digestibility were reported [13,35,68]. Kim et al. [30] showed increased in vitro DM degradability and insignificant effects on NDF and protein degradability with the administration of *A. saponaria*. In sheep, Lu and Jorgensen [41] stated that feeding a concentrate-based diet containing lucerne saponins enhanced the total tract digestibility of OM, cellulose, and hemicellulose. Liu et al. [35] showed that feeding ewes on diet supplemented with tea saponins at 2 g/ewe/day enhanced the in vivo digestibility of OM, NDF, and ADF. Recently, Taiwo et al. [68], using saponins extracted from fenugreek seed, noted increased DM, CP, NDF, and ADF digestibility in steers. The positive effects of saponins on nutrient digestibility indicate the ability of saponins to alter the site of digestion in the gastrointestinal tract [9], which possibly improves feed degradation in the rumen. Lu and Jorgensen [41] showed that lucerne saponins depressed the degradability of cellulose in the rumen, but increased cellulose and hemicellulose degradation in the hindgut. Additionally, saponins can increase the relative abundance of some fibrolytic microbes, such as *Bacterioidetes*, *Prevotella*, and *Prevotellaceae*, in pure cultures and in ruminal enrichment cultures [69]. Ridla et al. [13] observed that the positive effects of saponins on nutrient digestion were paralleled with low levels of saponins (<0.5% DM) whereas high levels decreased it. 

### 7.2. Microbial Protein Synthesis

As previously mentioned, saponins decrease the ruminal protozoal number. Lowering protozoal numbers increases protein synthesis by bacteria and slows ruminal protein turnover, resulting in the increased flow of bacterial N to the duodenum [9]. Saponins have an ability to partition nutrients such that a significant proportion of the digested substrate is used for microbial mass formation [9].

Widyarini et al. [25] showed increased microbial protein synthesis when *N. sativa* L. saponins were administered at 0.2%, 0.4%, and 0.6% DM through inhibiting protozoa (defaunation), which might increase the efficiency of microbial protein synthesis and protein flow to the duodenum [9]. In an experiment on heifers, Hristov et al. [40] showed unaffected intestinal flow of microbial N in heifers fed saponins from *Y. schidigera*. Hess et al. [47], however, reported improved flow of microbial protein to the duodenum with feeding sheep on diets containing *S. saponaria* fruit.

Saponin concentrations affect microbial protein synthesis. In an in vitro experiment, tea saponins at 8 mg against 200 mg mixture of corn meal and grass meal (1/1, w/w) in rumen fluid increased microbial protein synthesis [27]. Other experiments with *Y. schidigera* saponins at 100 mg sarsaponin/kg [57] or sapindus saponins [42] reported no effect on microbial protein synthesis. The observed discrepancy among experiments may be related to the different techniques of measuring microbial protein synthesis, dietary saponin concentrations, saponin type, and diet composition.

### 7.3. Ammonia-N and N Utilization

Ammonia, a product of ruminal feed fermentation and microbial lysis, is partly absorbed through the ruminal wall or utilized by microbes, which obtain 50–80% of their N requirements from the rumen ammonia-N pool. Decreasing ruminal protozoa numbers by saponins administration may reduce the proteolytic and deamination activities of protozoal origin [9]. Moreover, the inhibition of bacteria by saponins may reduce proteolysis and deamination. The ability of saponins to inhibit microbial urease causes reduction in ruminal ammonia concentrations. Wallace et al. [28] documented that the glycofractions of saponins bind ammonia, resulting in improved N use efficiency by maintaining adequate ruminal ammonia concentration. The glycofractions of saponins trap rumen ammonia when concentrations are high post-feeding, and then gradually release it for microbial protein synthesis when the concentrations decline.

The effect of saponins or plants containing saponins was extensively evaluated, with observed decreases in ruminal ammonia N concentrations in vitro [14,27,33] and in vivo [7,35]. However, the degree to which ammonia concentrations decreased differed among saponins from different sources [9]. Experiments [33] showed that saponins from *Y. schidigera* extract decreased ammonia efficiently compared to *Q. saponaria* extract, which may be related to the strong antiprotozoal properties of *Y. schidigera* compared with *Q. saponaria* saponins. Gunun et al. [7] observed lowered ruminal ammonia-N concentration 4 h postfeeding when they fed lactating cows on diets supplemented with *A. thwaitesianum* saponins. Jayanegara et al. [14] noted that saponins from *S. rarak* fruits were effective in mitigating ruminal ammonia production with more effects observed in a concentrate-based diet compared to a forage-based diet.

On the other hand, others observed unchanged ruminal ammonia-N with saponins of *S. saponaria* at 250 g/kg DM [47], *Y. schidigera* extract containing 10% saponins at 60 g/cow per day [39], and *A. saponaria* at 1% and 2% DM [30]. The concentration of saponins used may partially explain the weak effect. Moreover, the lowered protozoal number with saponins [47] may allow for increasing ruminal bacterial number as a result of the reduced predation and lysis of bacteria, which perhaps is the reason for saponin treatments not always resulting in decreased ruminal ammonia concentrations. Ruminal protozoa contribute about 10–40% of the total ruminal N. Inclusion of 10 g of *Y. schidigera* whole plant or 10 g *Q. saponaria* whole plant [33] had a marginal effect on ruminal ammonia-N concentrations. Kim et al. [30] noted the low effect of *A. saponaria* administration on in vitro ammonia-N concentrations. 

The rapid ruminal digestion of dietary protein causes the production of ammonia exceeding microbial needs, resulting in high excretion of urinary N. In the liver, excess ammonia is converted to urea and recycled through the ruminal wall, salivary secretion, and excreted in urine. Therefore, as previously noted, the reduced effect of saponins on rumen ammonia-N and urinary N excretion may increase fecal N excretion and the production of high-quality protein [24], which is available and well-absorbed in the lower gut [70]. Liu et al. [35] showed that supplementation of tea saponins at 2 g/ewe/day decreased daily fecal N output and urinary N, and increased N retention and N retention expressed as N intake. Saponins administration may enhance N absorption and retention due to improved salivary glycoproteins and digestive enzymes secretion, which possibly increase the regeneration of epithelial cells and mucus secretion in the digestive tract [70]. Gunun et al. [24] showed insignificant effects on N intake and total N excretion, decreased urinary N excretion, and increased N excreted in the feces, N absorption and retention, and N retention/N intake ratio when feeding goats with *T. chebula* (containing saponins at 94 g/kg DM) at 0, 8, 16, and 24 g/kg of total DM intake.

### 7.4. Volatile Fatty Acids

Volatile fatty acids (VFA), which are the main end products of feed fermentation in the rumen, are affected by saponins. Saponins form complexes with sterol moieties in the mucosal cells of membranes and thus affect the permeability of the intestinal cell. Additionally, saponins may decrease the mechanisms of active nutrient transport and enhance the small intestine membrane permeability to stimulate the uptake of materials to which the gut would normally be impermeable.

The effect of saponins on total and individual ruminal VFA concentrations differed, with no effects [24,25,33], adverse effects [36,38], and positive effects [30]. Responses of rumen VFA production to saponins depend on diet [9], application level, and rumen pH [71]. Cardozo et al. [71] observed that *Y. schidigera* saponins increased propionate and decreased acetate at a pH of 5.5, with no effects observed on VFA proportions at a pH of 7.0.

Extracts of saponins-containing *Acacia concinna* showed minimal effects on total VFA production, improved propionate, and reduced the acetate:propionate ratio. Widyarini et al. [25] observed insignificant effects with the administration of saponins from *N. sativa* L. at 0.2%, 0.4%, and 0.6% (DM basis) on total or individual VFA concentrations. Recently, Gunun et al. [24] reported unchanged total and individual VFA production in goats fed *T. chebula*.

Lucerne saponins at 4% of DM intake [41] and *Y. schidigera* extract at 25 and 50 g/d in Holstein cow diets decreased total VFA concentrations [36]. Hussain and Cheeke [38] observed that the administration of *Y. schidigera* saponins at 250 mg/kg feed to steers fed diets based on concentrate or roughage numerically decreased propionate concentration. The negative impact of saponins on ruminal fermentation is probably due to the use of high levels of saponins [9]. On the other hand, *A. saponaria* administration induced higher total and individual VFA concentrations [30].

In some experiments [7,31], enhanced propionate or reduced acetate proportions were observed with saponins from different plants. Santoso et al. [31] showed that an extract of saponins-containing *Biophytum* offered to goats at 160, 239 and 319 ml twice daily, corresponding to 13, 19.5 and 26 mg of saponin/kg body weight, increased propionate concentration. Gunun et al. [7] observed a decreased acetate proportion and an improved propionate proportion in lactating cows fed diets supplemented with saponins-containing *A. thwaitesianum*. The increased propionate, and lowered acetate and butyrate may be due to the ability of saponins to decrease the number of ruminal protozoa, which majorly produce acetate and butyrate as the main end products of their fermentation [9].

Therefore, decreasing protozoal populations through the use of saponins may improve propionate proportions and the ratio of propionate to acetate. Additionally, saponins stimulate the growth of *S. ruminantium* [48], a propionate producer.

### 7.5. Gas and Methane Productions

Gas production is an important indicator of feed nutritive value. The ruminal digestion of feed by ruminal microbes produces gases, mainly methane, carbon dioxide, and hydrogen. It is expected that gas production will be affected by the administration of saponins or saponins-containing plants due to their effects on nutrient digestion. Jayanegara et al. [14] noted that the addition of saponins from *S. rarak* fruits at 0.5, 1, 1.5 and 2 mg/ml medium to two diets (high-forage or high-concentrate) did not affect gas production. Recently, Kim et al. [30] showed that *A. saponaria* (containing saponins at 37.26 and 45.65 mg/g) at 1 and 2% increased gas production without affecting theoretical maximum gas production or fractional rate of gas production compared to the control treatment.

Ruminal methane is produced by methanogenic archaea as a byproduct, which contributes to feed energy loss and adds to the greenhouse effect. Propionate formation competes with methane formation for hydrogen availability; therefore, increasing propionate production decreases methane formation [72]. The effects of saponins on methanogenic archaea numbers and methane production are controlled by several factors [73]. Saponins may reduce the activity of methane-producing *mcrA* genes, methane production rates, and methanogenic archaea numbers and/or activity [26,42]. Many methanogens exist in both ecto- and endosymbiotic relationships with protozoa and are responsible for about 37% of ruminal methanogenesis [42]. Each single protozoan cell may contain 10^3^–10^4^ methanogens pre-feeding, which decreases to one to ten methanogens post-feeding.

The engulfment of rumen methanogens by protozoa increases the populations of methanogen that are not in association with protozoa. The higher suppression of protozoa is not always linked to a higher reduction of methanogens and methanogenesis. In an experiment with *A. saponaria* administration at 37.26 and 45.65 mg/g, Kim et al. [30] observed a lowered protozoal number and an increased archaeal number without affecting methane production. 

Hess et al. [47] confirmed these effects in an in vivo experiment where lambs were supplemented with *S. saponaria* at 0.6 g/kg BW^0.75^. Saponins of tea at 67 mg/L and 133 mg/L deceased in vitro methane production by 13 and 22%, respectively, but increasing the dose to 200 and 267 mg/L did not mitigate methane production further [27]. Linear decreases in methane production were observed with increasing levels of *Y. schidigera* saponins at 1.2, 1.8, 2.4, and 3.2 g/L in an incubation medium, using potato starch, maize starch, and hay-concentrate as substrates [59]. Decreased in vitro methane production was observed with whole *Y. schidigera* saponins at 0.023, 0.046, and 0.069 g/L [33]. Wang et al. [74] showed that the inclusion of *Gynostemma pentaphyllum* (98% gynosaponin) saponins at 50, 100, or 200 mg/L mediums decreased methane production. Widyarini et al. [25] showed a reduced in vitro methane production with the administration of *N. sativa* L. saponins at 0.2%, 0.4%, and 0.6% DM. Goel and Makkar [75] observed decreased CH_4_ production (by 34–48%)with the addition of *Achyranthus aspara*, *Tribulus terrestris*, and *Albizia lebbeck* saponin extracts at 3, 6, or 9% dietary DM. Jayanegara et al. [14] observed that saponins from *S. sesban* inhibited methane production effectively in diets based on concentrates compared with diets based on roughage. *S. saponaria* saponins at 120 mg/g were more noticeable in defaunated (29%) rumen fluid relative to the faunated fluid (14%). Moreover, diet affects the response of methane production to saponins. 

Klita et al. [67] showed that the intra-ruminal administration of lucerne root saponins at up to 4% of DM intake did not affect methane production, linearly decreased the protozoal counts, inhibited the ruminal motility, and increased intestinal motility in sheep. Klita et al. [67] conjectured that epithelial receptors in the luminal epithelium might be involved, possibly due to the interactions of saponins with the sterols in biological membranes. Patra and Saxena [9] explained the mechanism by which saponins affect intestinal motility through the modulation of cells for pace making in the intestinal muscles, which produce rhythmic oscillations in the membrane potential; this modulation is mediated through non-selective cation channels and intracellular Ca^2+^ mobilization in a protein kinase C-independent manner, as shown with ginseng saponins. Recently, Kim et al. [30] observed unchanged methane production and increased ruminal archaea abundance with the administration of *A. saponaria* at 1 and 2% DM in vitro.

Some experiments indicated that saponins may also increase methane production, possibly due to increased ruminal bacterial and fungal populations, which perhaps increase nutrient digestibility, especially fibers [73]. The effect of saponins on the digesta passage rate (inverse relationship), which increases with increased fiber digestion, may be another justification for enhanced methane production in some experiments [41,67]. Guyader et al. [73] noted enhanced methane emissions (g/kg of dry matter intake) by 14% in lactating dairy cows fed a basal diet supplemented with 0.52% DM tea saponin.

## 8. Effects of Saponins on Blood Parameters

The main parameter in blood that may be affected by saponins is urea. Blood urea N is an index of the metabolic status of amino acids in ruminants. Saponins of *Y. schidigera* extract have the ability to bind ammonia, causing ammonia to be slowly released, which expectedly affects blood ammonia or urea levels [28]. Reducing amino acid deamination and ruminal bacterial proteolysis decreases ammonia production and increases microbial protein flow and amino acids availability for intestinal absorption.

Unchanged plasma ammonia or urea concentrations when feeding with a saponins-containing *Y. schidigera* extract were observed in steers at 250 mg kg feed [38] and heifers at 20 or 60 g/d [40] fed high-roughage or high-concentrate diets. Śliwiński et al. [76] showed unaffected hematocrit or hemoglobin concentrations in dairy cows supplemented with saponins from *Y. schidigera* at 0.1 g/kg. Others [47] showed that sheep fed a concentrate feed mixture containing 250 g *S. saponaria* fruit per kg had lowered plasma urea, indicating less ruminal ammonia absorption. Gunun et al. [7] showed that feeding *Antidesma* to lactating cows at different levels, as a source of saponins, had no effect on blood urea-N, hemoglobin, hematocrit, white blood cells, lymphocytes, neutrophils, monocytes, and eosinophils.

Abdullah and Al-Galbi [77] showed that feeding saponins from tea leaves at 180 mg/kg DM increased blood glucose and protein concentrations without affecting cholesterol. Recently, Taiwo et al. [68] observed lowered concentrations of blood urea and glucose when feeding with a saponins-containing fenugreek seed extract to steers. Moheghi et al. [32] reported that feeding with a basal diet consisting of forage (30%), concentrates (70%), and saponins (150 mg/kg DM) to Baluchi lambs had a marginal effect on the concentrations of glucose, blood urea N, total protein, triglycerides, cholesterol, packed cell volume, total protein, and hemoglobin, but increased the count of total white blood cells.

## 9. Effects of Saponins on Animal Performance

The effect of feeding saponins or plants containing saponins on animal performance varied among experiments depending on the diets, and the source and levels of saponins involved (Table 2). As previously noted, the antiprotozoal (defaunation) effects of saponins are expected to increase ruminant performance, especially those on a low-protein diet. Moreover, it was stated that the effects of saponins are diet-dependent. Animals requiring high protein but placed on low-true-protein diets inadequate in energy benefit more from the antiprotozoal effects of saponins.

### 9.1. Growth Performance

Improved animal growth performance through feeding with saponins was observed in many experiments, as a result of the increased intestinal absorption of amino acids due to decreased protozoal numbers [9]. Feeding with a diet based on maize silage and supplemented with 150 mg sarsaponin daily to steers increased daily gain during the first 28 d of an experiment, without affecting long-term growth at 62 d, which was probably due to adaptation to saponins. Abdullah and Al-Galbi [77] showed that feeding saponins from tea leaves at 180 mg/kg DM improved total gain, from 15.0 kg in control to 17.5 kg in treatment, and a feed-to-gain ratio.

Contrarily, feeding saponins to animals fed concentrated diets showed weak effects on growth performance. Hussain and Cheeke [38] observed minimal effects on the daily gain of sheep supplemented with 250 mg/kg *Y. schidigera* extract and fed a mixed diet containing 45% hay, 50% rolled barley, and 5% soyabean meal. Supplementation with *Y. schidigera* powder at 150 mg/kg to male lambs fed a diet containing 90% concentrates marginally affected their weight gain [78]. Görgülü et al. [78] noted that supplementing the diet (90% concentrates) of male lambs with 150 mg/kg *Y. schidigera* powder did not affect their weight gain. Recently, Moheghi et al. [32] observed unchanged daily gains, final body weights, and feed conversion when feeding crude saponins to Baluchi lambs at 150 mg/kg DM feed. Moreover, they did not observe significant differences in fatty acids profiles (concentrations of the saturated and unsaturated fatty acids, and their ratios) of the *Longissimus thoracis* muscle.

### 9.2. Lactation Performance

The influence of saponins on feed intake showed weak effects in many experiments [40]. Other experiments showed slight decreases in feed consumption ranging from 2 to 6% [36,39], resulting in lowered milk production efficiency. However, some experiments showed increased feed intake when feeding with saponins [68]. It should be emphasized that the weak effect of saponins on milk production and composition may be related to using low doses of saponins. Low levels of saponins did not affect rumen fermentation and protozoa numbers, resulting in a low effect on milk yield or composition, as previously noted [39].

Lovett et al. [36] and Holtshausen et al. [33] reported that feeding with a *Y. schidigera* extract at 25 and 50 g/d lowered feed consumption without any effect on milk yields, resulting in increased milk conversion efficiency. The situation differs in dairy animals compared to growing animals (i.e., meat production), even if the supplementation of saponins was in diets containing low protein. Śliwiński et al. [76] showed that feeding diets with <10% protein and supplementation with saponins from *Y. schidigera* at 0.1 g/kg had no effect on milk yield in cows. Moreover, supplementation of *Y. schidigera* or *Q. saponaria* powder at 10 g/kg of DM to lactating animals fed diets containing 10 to 20% crude protein did not affect milk yields or composition [33,39]. In another experiment, Gunun et al. [7] observed that feeding saponin containing *A. thwaitesianum* at different levels (9.8, 19.6 and 29.4 g/cow/d) had no effect on milk yield and composition or on the somatic cell counts in lactating cows. However, Guyader et al. [73] showed that the supplementation of lactating dairy cows’ basal diet with 0.52% tea saponin decreased milk production, intake, and feed efficiency by 18, 12, and 8%, respectively.

## 10. Saponins as Anthelmintics

Saponins may be used to control internal parasites in ruminants. Saponins can inhibit the activity of proteases, lipases, and chitinases, which degrade the egg membranes that are important for the hatching process of nematode eggs [79]. Changes in the activities of these enzymes disrupt the egg-hatching process, leading to the destruction of infectious worms [79]. Botura et al. [80] stated that *Agave sisalana* (containing hecogenin and tigogenin) at 1.7 g/goat/day decreased a fecal egg population by about 50.3%. Moreover, Botura et al. [79] showed that the ethanolic extract of *Phytolacca icosandra* destroyed *Haemonchus contortus* eggs and larvae, while *Agave sisalina* attacked nematodes in the gastrointestinal tract of animals.

## 11. Conclusions

Saponins and plants containing saponins (whole plants or extracts) exhibit some beneficial effects as a feed or as feed additives for ruminants. High concentrations of saponins may act as natural rumen manipulators to modify the composition and fermentation of ruminal microbial populations, which may change rumen metabolism positively or negatively. The main important effect of saponins is ruminal defaunation. Saponins suppress ciliate protozoa and may thus enhance microbial protein synthesis efficiency by decreasing microbial protein turnover and the duodenal flow of protein. The antiprotozoal effect of saponins may inhibit methanogenesis by reducing the activities of ruminal methanogens. Saponins affect ammonia adsorption and modulate the ruminal passage of digesta, causing altered rumen metabolism with negligible physiological responses compared to the microbiological effects. Due to the beneficial effect of saponins on N metabolism, their use to overcome problems associated with inefficient/poor N retention and utilization in ruminants can be recommended. The effects of saponins on ruminal microflora and fermentation depend on the types and levels of saponins, diet composition, and on microbiota populations and adaptation to saponins. Different forms of saponins produce different activities. One of the factors influencing the benefits of saponins administration in ruminants is the identification of the main bioactive/potent saponins that can specifically inhibit protozoa and methanogens. The majority of saponins are considered quite safe and beneficial, but certain types may be poisonous to animals, and there is no clear reason why some saponins are beneficial while others are toxic. The adaptability of ruminal microbes to saponins after long-term use is an issue that needs more evaluation. Saponins are more effective in enhancing the performance of animals fed high-roughage diets; therefore, they would be beneficial for smallholder livestock farmers in developing countries.

## Figures and Tables

**Figure 1 vetsci-10-00450-f001:**
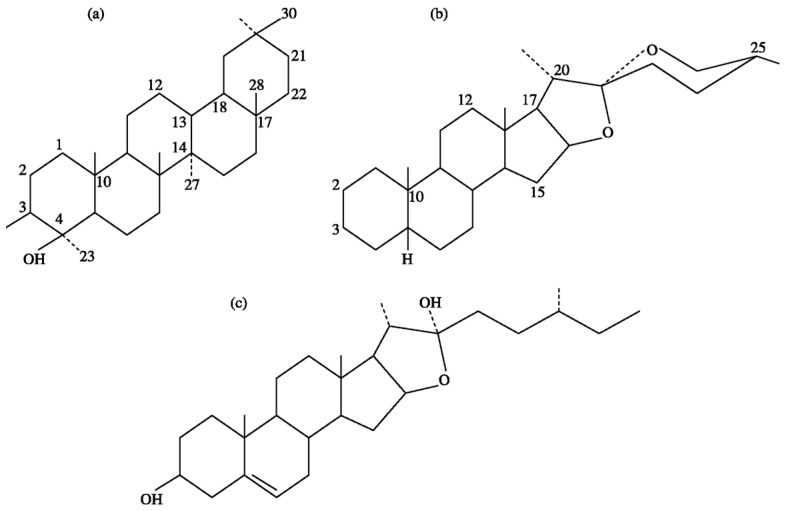
Chemical structures of sapogenins: (**a**) triterpenoid; (**b**) steroids; and (**c**) furostanol.

**Table 1 vetsci-10-00450-t001:** Effects ^1^ of saponins or saponins-containing plants on ruminal fermentation.

Saponin Source ^2^	Animal	Dose	Feed (Forage:Concentrate Ratio)	Protozoa	Ammonia	Volatile Fatty Acid	Acetate: Propionate	Microbial Protein Synthesis	Methane	Reference
*Aloe saponaria*	Holstein steers	1 and 2% of total diet	30.5:69.5	NR	=	+	+	NR	=	[30]
*Antidesma thwaitesianum*	Cow	9.8, 19.6 and 29.4 g/cow/d	0:100	−	−	=	−	NR	−	[7]
*Biophytum petersianum*	Goats	13 mg/kg BW	70:30	−	−	−	−	=	NR	[31]
*Biophytum petersianum*	Goats	26 mg/kg BW	70:30	−	−	=	−	+	NR	[31]
*Biophytum petersianum*	Goats	19.5 mg/kg BW	70:30	−	−	−	−	+	NR	[31]
Crude saponins	Lambs	150 mg/kg DM feed	30:70	NR	=	NR	NR	NR	NR	[32]
*Medicago sativa* L.	Cow	2.4 mg	60:40	−	−	NR	NR	NR	−	[10]
*Nigella sativa*	Cattle	0%, 0.2%, 0.4%, and 0.6% saponin	100:0	+	−	=	=	−	−	[25]
*Quillaja saponaria* plant	Dairy cows	10	51:49	=	=	=	=	NR	=	[33]
*Sapindus rarak*	Cow	2 mg/mL medium	30:70	−	−	+	−	NR	−	[14]
*Sapindus Saponaria* fruit	Sheep	5 g/kg BW^0.75^	67:33	−	=	+	−	NR	−	[26]
Sarsaponin	Dairy cows	0.2, 0.41, 0.62	36:64	=	−	=	=	NR	NR	[34]
Tea	Sheep	2 g/ewe/day	68.7:31.3	−	−	+	−	=	NR	[35]
*Terminalia chebula*	Goat	0, 8, 16, and 24 g/kg of total DM intake	50:50	NR	−	=	=	NR	NR	[24]
*Yucca schidigera* extract	Steers	2.56	63:37	−	NR	−	=	NR	NR	[36]
*Yucca schidigera* extract	Sheep	0.13	75:25	NR	−	+	=	NR	−	[37]
*Yucca schidigera* extract	Steers	0.075	92:8, 96:4, 45:55, 48:52	NR	=	=	=	NR	NR	[38]
*Yucca schidigera* extract	Dairy cows	2.8	40:60	=	=	=	=	NR	NR	[39]
*Yucca schidigera* plant	Heifers	1.96 and 5.83	61:39	−	=	=	−	=	NR	[40]
*Yucca schidigera* plant	Dairy cows	10	51:49	=	−	=	=	NR	=	[33]

NR means not reported; − means decrease; + means increase; = means no effect. ^1^ Relative to control. ^2^ Values in parentheses are the saponin content (%) in extracts or plants.

**Table 2 vetsci-10-00450-t002:** Effects ^1^ of saponins or saponins-containing plants on animal performance.

Saponin Source ^2^	Animal	Dose	Feed (Forage: Concentrate Ratio)	DM Intake	Average Daily Gain	Milk Production	Reference
*Antidesma thwaitesianum*	Cow	9.8, 19.6, and 29.4 g/cow/d	0:100	=	NR	=	[7]
Crude saponins	Lambs	150 mg/kg DM	30:70	=	=	NR	[32]
*Quillaja saponaria* plant	Dairy cows	10	51:49	+	=	=	[33]
Sapin*dus Saponaria* fruit	Sheep	5 g/kg BW^0.75^	67:33	=	+	NR	[26]
Sarsaponin	Dairy cows	0.2, 0.41, and 0.62	36:64	=	=	=	[34]
Tea leaves	Lambs	180 mg/kg DM feed	NR	=	+	NR	[77]
*Yucca schidigera* extract	Dairy cows	1.46 and 3.1	63:37	=	NR	=	[36]
*Yucca schidigera* extract	Steers	1.25 and 2.56	63:37	=	NR	=	[36]
*Yucca schidigera* extract	Sheep	0.13	75:25	=	NR	NR	[37]
*Yucca schidigera* extract	Dairy cows	2.8	40:60	NR	NR	=	[39]
*Yucca schidigera* plant	Dairy cows	10	51:49	+	=	=	[33]

NR means not reported; + means increase; = means no effect. ^1^ Relative to control. ^2^ Values in parentheses are the saponin content (%) in extracts or plants.

## Data Availability

Not applicable.

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
