# Peer review of "A Review of Effect of Saponins on Ruminal Fermentation, Health and Performance of Ruminants"

_vetsci, 2023, doi:10.3390/vetsci10070450_

Round 1

Reviewer 1 Report

I consider the review on saponins to be quite complete and I would only kindly suggest to the authors that they prepare another paper with a meta-analysis of saponins, which will contribute significantly to work with this phytogenetic

Author Response

The author thanks the reviewer for his suggestion. A meta-analysis manuscript on saponins will be prepared. I hope the present revision is acceptable for publishing in Veterinary Sciences.

Reviewer 2 Report

The MS is a review about the Effect of Saponins on Ruminal Fermentation, Health and Performance of Ruminants. The logic is good and the presentation is well organised. 

However, there is one point in line L110.  please control the value 50%. 

extensive english corrections are needed: such as in line 373, that is extra. in line 682, maybe the author means 10 or 20% lower? 

Author Response

 Authors: The author thanks the reviewer for his suggestion. All issues raised by the reviewer have been addressed. I hope the present revision is acceptable for publishing in Veterinary Sciences.

However, there is one point in line L110.  please control the value 50%. 

Authors: Revised to read 25% to 50% (https://doi.org/10.1038/s41396-021-01170-y, https://doi.org/10.3389/fmicb.2020.00720, https://doi.org/10.1017/S1751731118001957).

Comments on the Quality of English Language extensive english corrections are needed: such as in line 373, that is extra. in line 682, maybe the author means 10 or 20% lower? 

Authors: The manuscript has been revised by Prof. Olurotimi Olafadehan (oaolafadehan@yahoo.com) Professor of Ruminant and Dairy Production, Department of Animal Science, University of Abuja, Abuja, Nigeria. I hope the present revision is acceptable for publishing in Veterinary Sciences.

Reviewer 3 Report

This review titled " Review of Effect of Saponins on Ruminal Fermentation, Health and Performance of Ruminants" by Ahmed E. Kholif is more interesting and well written. I have no suggestions for the author and send him my compliments.

Author Response

The author thanks the reviewer for his compliments. I hope the present revision is acceptable for publishing in Veterinary Sciences.